# Peer-driven intervention to help patients resume CPAP therapy following discontinuation: a multicentre, randomised clinical trial with patient involvement

Raymond Merle,[1] Christophe Pison ,[1,2] Sophie Logerot,[3] Chrystèle Deschaux,[3] Nathalie Arnol,[3] Matthieu Roustit ,[1,4] Renaud Tamisier,[2,5] Jean Louis Pépin,[1,4] Jean Christian Borel [3]

JLP and JCB are joint senior authors.

For numbered affiliations see end of article.

**Correspondence to**
Professor Christophe Pison;
CPison@chu-grenoble.fr

## ABSTRACT

**Introduction** Obstructive sleep apnoea syndrome (OSAS) is one of the most common chronic diseases. It may be associated with symptoms of excessive daytime sleepiness and neurocognitive and cardiovascular complications. First line therapy for OSAS involves home continuous positive airway pressure (CPAP), however, nearly half of patients do not adhere with this treatment over the long term. Cognitive-behavioural interventions that include health professionals and patient and public involvement are increasingly advocated in the fields of education and research. We hypothesise that a peer-driven intervention could help patients with OSAS to resume CPAP use after discontinuation.

**Methods and analysis** We have designed a prospective, multicentre randomised, controlled trial that will be coconducted by health professionals, a home provider of CPAP and patients as experts or peers or participants. The primary aim is to evaluate the impact of a 6-month, peer-driven intervention to promote the resumption of CPAP after discontinuation. We anticipate that 20% of patients in the intervention group will reuse CPAP as compared with 6% in control group, thus, 104 patients must be included in each group. The secondary aims are (1) to evaluate the impact of the peer-driven intervention on adherence to CPAP compared with the control group (mean adherence and percentage of nights with at least 4 hours' use/night for 70% of nights); (2) to determine factors associated with resumption of CPAP; (3) to assess patient satisfaction with the peer-driven intervention at 6 months; (4) to evaluate the feasibility and the execution of the peer-driven intervention and peer satisfaction. Adult outpatients with an established diagnosis of severe OSA (Apnoea-Hypopnoea Index >30 events/hour) that have stopped using CPAP within 4–12 months after initiation will be recruited. The peers who will perform the intervention will be patients with OSAS treated with CPAP with good adherence (at least 4 hours/night, 70% of nights) and trained in motivational enhancement and cognitive-behavioural therapies. Trained peers will conduct three interviews within 6 months with participants.

**Strengths and limitations of this study**

► Patient involvement from the beginning of the setup of this trial.
► There is a rationale supporting the use of motivational enhancement and cognitive-behavioural therapies performed by peers to promote continuous positive airway pressure resumption.
► Our team has experience in patient and public involvement from work undertaken at the Grenoble Alpes University Department of Patients.
► Challenges are to train enough peers with homogeneous skills.

**Ethics and dissemination** Ethical approval has been obtained from the French Regional Ethics Committee CPP Ouest II-Angers, (IRB 21.02.25.68606 (2021/2025)). All participants will sign written informed consent. The results will be presented at conferences and published in peer-reviewed journals as well as public media.
**Trial registration number** NCT04538274

## INTRODUCTION

Obstructive sleep apnoea syndrome (OSAS) is one of the most common chronic diseases. It is characterised by recurrent episodes of upper airway collapse during sleep, and may or may not be associated with symptoms of excessive daytime sleepiness (EDS) and neurocognitive and cardiovascular complications.[1] Twelve million adults aged between 30 and 69 years may have moderate to severe OSAS in France, based on an Apnoea–Hypopnoea Index (AHI) threshold value of 15 or more events per hour of sleep.[2] The risks associated with the disease can be severe, for example, individuals with untreated OSAS have a three times greater risk of motor vehicle accidents than the general population.[3] OSAS is also

BMJ

associated with an increased risk of cardiovascular disease, diabetes and glucose dysregulation,[4] independent from obesity.[5]

The first line therapy for OSAS is continuous positive airway pressure (CPAP).[1 6 7] CPAP has been shown to effectively reduce EDS and to improve daily functioning, cognitive function, mood and quality of life.[3 6] The use of CPAP also reduces traffic accidents[7] and other work-related injuries, and improves work productivity.[8] Although CPAP therapies are highly effective in normalising AHI and reducing symptoms in symptomatic patients, treatment success is limited by long-term non-adherence in nearly half of patients.[9] Technical progress in the systems and interfaces (soundproofing, improved masks, humidification, pressure modulation, etc) have unfortunately not been sufficient to improve compliance.[10 11] Equally, the effect sizes of telemedicine approaches are not as large as what has been achieved with the use of behavioural therapies, and the impacts on patient and provider satisfaction and cost-effectiveness are not yet clear.[12–15]

Non-adherence is related to users' profiles, their representations of OSAS and the benefits they experience from CPAP.[12 16 17] This is why cognitive-behavioural and motivation enhancement therapies conducted by health professionals could be effective in ensuring adherence to CPAP. A Cochrane review in 2014 showed that there is a low level of evidence that such interventions increase CPAP use (by 1.44 hours per night in six studies; n=584) and increase the number of participants who used their devices for longer than 4 hours per night (from 28% to 47% in three studies; n=358).[18] More robust studies are thus needed to increase the level of evidence regarding these types of interventions.

In addition, patient and public involvement (PPI) is more and more advocated in the fields of health education and research.[19–25] Nevertheless, the efficacy of PPI remains to be demonstrated.[26] To our knowledge, only one previous pilot study in 39 patients showed that one-to-one peer support at CPAP initiation was feasible and generated high patient satisfaction. However, the study was not powerful enough to demonstrate effectiveness in terms of adherence to CPAP.[18 27] The data from the study, are, however, useful for designing further studies.

The aim of this adequately powered randomised clinical trial is therefore to assess the role of trained patient involvement (PI) representatives to help patients with OSAS to restart CPAP after discontinuation.

## METHODS AND ANALYSIS
### Study design
This is a prospective, multicentre, randomised controlled trial that will be coconducted by health professionals, a CPAP home provider and patients as experts or peers or participants. After signing a consent form, patients' participants will be randomised 1.1 to the intervention group with peers or the control group. Nota bene: the peers involved in the conduct of the study will sign a confidentiality agreement of non-divulgation of the information exchanged with the participants.

## Objectives
### Primary research aim
The primary aim is to evaluate the impact of a 6-month intervention involving trained PI representatives to promote the resumption of CPAP in patients who have discontinued its use.

### Primary research outcome
Resumption of CPAP is defined as the medical prescription and the setting up of a new CPAP device at home by the homecare provider.

### Secondary research aims
1. To evaluate the impact of the peer-driven intervention on adherence to CPAP by comparing adherence with the control group (mean adherence and % of nights with at least 4 hours' use /night for 70% of nights).
2. To determine the factors associated with the resumption of CPAP treatment.
3. To assess the satisfaction of the intervention group with the peer-driven intervention at 6 months.
4. To evaluate the feasibility and the execution of the peer-driven intervention and the satisfaction of peers after the interviews conducted.

### Secondary research outcomes
1. Average adherence to CPAP will be measured from data recorded by the built-in software of the CPAP devices (via tele monitoring or retrieved by a home technician) for 1 month after the final consultation.
2. The relationship between the variables below and a positive response to the peers intervention (defined by a restart of CPAP treatment) will be analysed: age, gender, body mass index, marital status and number of young children (<10 years) education level, socio-professional status, fragility and social precariousness (using the EPICES score), smoking and alcohol use, comorbidities (using Charlson score), history of OSAS (date of diagnosis of OSAS, baseline AHI), observance to treatments (Girerd score), date and reason for stopping CPAP and EDS score (using the Epworth Sleepiness Scale). To determine patient profiles, their representations of OSAS, their experiences with CPAP and their knowledge and confidence to manage their health, three questionnaires will be completed at inclusion (M0) and at the 6-month follow-up (M6): the Functional Outcomes of Sleep Questionnaire a disease-specific quality of life questionnaire,[28] the patient activation measure a measure that assesses patient knowledge, skill and confidence for self-management[29] and the Self-Efficacy Measure for Sleep Apnoea[30 31] a tool with strong psychometric properties that identifies patient perceptions that may indicate those most likely not to adhere to treatment.
3. The satisfaction of participating patients with the PI intervention and the satisfaction of PI representatives

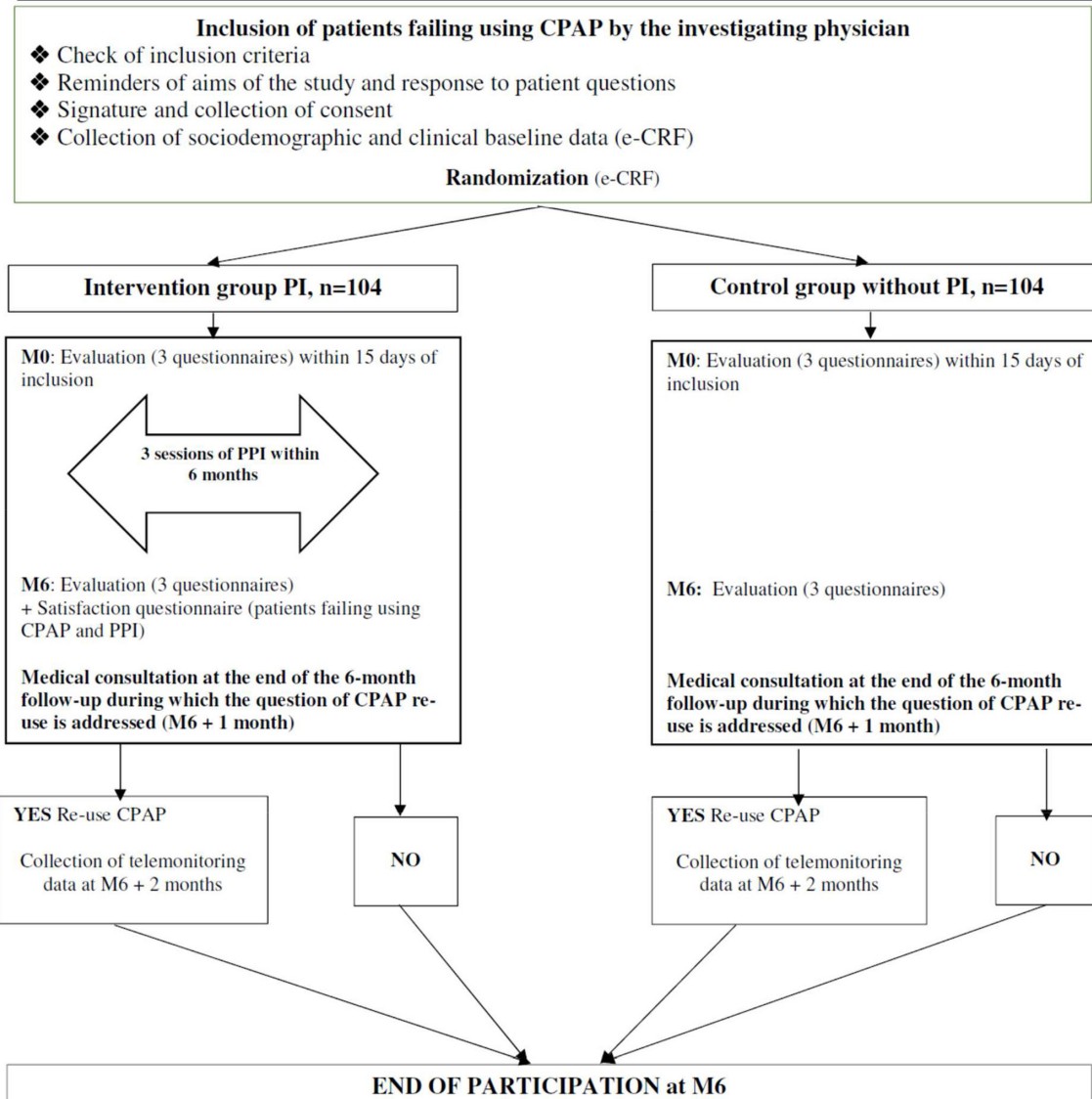

**Figure 1** Study design. CPAP, continuous positive airway pressure; PI, patient involvement; PPI, patient and public involvement. e-CRF, electronic-Clinical Record Form.

will be measured on a 4-point Likert scale: very dissatisfied, dissatisfied, satisfied, very satisfied.

4. The feasibility and the execution of the three interviews will be assessed by the number of interviews carried out in their entirety and the average duration of each interview (in minutes).

All information will be collected in secure electronic medical records in accordance with the requirements of general data protection regulation.

## Patients

Adults with an established diagnosis of severe OSAS (AHI >30 events/hour) who have discontinued CPAP by returning their device to the homecare provider within 4–12 months after CPAP initiation will be recruited according to the study flow chart depicted in figure 1 (table 1).

## Interventions

### Recruitment and training of PI representatives

PI representatives will be recruited from the investigators clinics. To be recruited as a PI representative, patients should:

► Have used home CPAP for at least 1 year.
► Have a CPAP adherence of at least 4 hour/night for 70% of nights.
► Express their motivation in participating in a training and orientation session conducted by research staff and including expert patients from the Grenoble Alpes University Department of Patients (DUP GA).[23]
► Accept to conduct three motivational sessions by videoconference meetings of 45–60 min duration with 5–8 patients within 6 months after each patient's inclusion.

**Table 1** Inclusion and exclusion criteria

| Inclusion criteria | Exclusion criteria |
|---|---|
| ▶ Over 18 years old<br>▶ Diagnosed with of severe OSA (AHI ≥30 events/hour)<br>▶ Discontinuation of CPAP 4–12 months after initiation, despite the interventions of health professionals and provider, and having stopped their CPAP treatment no later than 1 year prior to their inclusion<br>▶ Followed by the home healthcare provider AGIRa*Dom*.<br>▶ Access to a computer and/or tablet and an internet connection.<br>▶ Oral and written French.<br>▶ Able to provide written informed consent.<br>▶ Affiliated to social security or beneficiary of such a scheme. | ▶ CPAP cessation due to a resolution of the OSAS (eg, weight loss after bariatric surgery) or another pathology that prevents the continuation of treatment (eg, ENT surgery).<br>▶ Severe and/or unstable comorbidity that required hospitalisation for decompensation in the previous year (heart, kidney, respiratory, liver, psychiatric or other insufficiency).<br>▶ Central sleep apnoea index above 20% of AHI at the time of diagnosis<br>▶ Patient being treated with a mandibular advancement orthosis.<br>▶ Lack of availability (eg, night worker or patient who travels frequently).<br>▶ Current participation in, participation in the month prior to inclusion in another clinical intervention research study that may impact the study: this impact is left to the investigator's discretion.<br>▶ Referred to in Articles L1121-5 to L1121-8 of the socio-professional class (corresponds to all protected persons: pregnant woman, breastfeeding mother, person deprived of liberty by judicial or administrative decision, person subject to a legal protection measure). |

AHI, Apnoea–Hypopnoea Index; CPAP, continuous positive airway pressure; ENT, ear, nose, throat; OSA, obstructive sleep apnoea; OSAS, obstructive sleep apnoea syndrome.

Patients with any major psychiatric illness, shift workers or frequent out of town travellers will not be recruited as peers.

Peers will be trained during a three half-days interactive session organised by DUP GA, with experts in patient therapeutic education and communication, and investigators.[23] Peers will be taught how to interact with the patients recruited in the study: the aim is for them to share their experiences but not to provide any medical advice.

### Description of the intervention

Trained peers will meet patients randomised into the intervention group by videoconference. Each PI representative will be allocated 5–8 patients. They will conduct three face-to-face motivational sessions, each of 45–60 min duration, over a 6-month period based on the principle of motivational enhancement and cognitive-behavioural therapies.[11 13] The content of the first session is designed to identify and understand the underlying reasons for stopping CPAP treatment and to identify difficulties encountered by the patient (advantages and disadvantages of CPAP treatment). The aim of the second session will be for the patient to define his/her objectives and priorities. During the last session, will be discussed to strengthen the motivation to change and how to plan for it. The peers will receive €100 per patient for the three interviews.

In the control group, patients will be informed, at inclusion, that they can have a visit with a physician investigator at any time to resume treatment if they wish, as is usual practice. At the end of the 6-month follow-up period, all patients in both groups will have a consultation with their physician who will suggest they resume CPAP treatment. This visit may take place earlier if the patient wishes to resume CPAP treatment before the end of the follow-up period. We planned to start inclusions by November 2021 and end the study by December 2023.

### Statistical analysis

#### Sample size

We hypothesise that 20% of patients allocated to the intervention group will reuse CPAP 6 months as compared with 6% of patients in the control group. A two group $\chi^2$ test with a 5% two-sided significance level will have 80% power to detect such difference between the two groups when the sample size in each group is 90 (nQuery V.8, Statistical Solutions, Cork, Ireland). In order to take into account a possible drop-outs and to comply with the intention-to-treat principle, we will inflate the sample size by a factor of 15%.[32] We, thus, plan to include 104 patients per group (ie, 208 patients in total). Fifteen patient peers will be involved.

#### Feasibility and recruitment

The home care provider, *AGIR* à dom. follows more than 20 000 patients with OSAS who use CPAP in the south of France. In 2018, out of 3281 patients who started CPAP within the study area (Isère, Savoie and Haute-Savoie), 365 discontinued it between 4 and 12 months postinitiation and 6% resumed use within 6 months after discontinuation.

## Randomisation

After consent, randomisation will be performed by a centralised computer software for each investigating centre. It will be stratified on the centre.

## Statistical analysis plan

### Descriptive analyses

Continuous variables will be expressed as medians (25th/75th percentiles) or means (SD) depending on normality which will be assessed with the Shapiro-Wilk test. Categorical variables will be reported as absolute numbers and percentages for both groups. Baseline comparisons between groups will be made using a Student's t-test or Mann-Whitney U test, depending on the distribution. For discrete variables, a $\chi^2$ test will be used. If significant differences are observed between arms, analysis of variance and multivariable regression will be performed. In the case of missing data, an imputation strategy will be applied according to the percentage of missing values. Data management and statistical analyses will be performed using SAS, V.9.4, SAS Institute.

### Primary outcome analysis

The impact of the PI intervention on the resumption of CPAP treatment will be studied by comparing the resumption of CPAP in the two arms, using a $\chi^2$ test. To take into account a possible centre effect, a second analysis will be carried out using a conditional logistic regression stratified by the centre; the intervention or control arm will be considered as the dependent variable.

### Secondary outcomes analyses

Mean CPAP compliance 1 month after resumption of CPAP will be analysed using a mixed linear model (fixed factor: randomisation arm (intervention vs control), random factor: centre). Comparison of the probability of resuming CPAP with an average compliance of at least 4 hours/night, 70% of nights between the intervention and control groups will be analysed using a conditional logistic regression, stratified by centre. All analyses will be performed as intention-to-treat and then a sensitivity analysis will also be performed per protocol (patients who have not resumed treatment will be considered to have zero adherence).

The association between resumption of CPAP and the sociodemographic parameters, clinical data and the scores of the three questionnaires will be studied by conditional logistic regression models stratified by centre, and adjusted by arm (intervention vs control).

In the intervention arm, descriptive statistics will be presented on the satisfaction as well as on the number of interviews carried out and their average duration.

## Ethics

The study will be conducted in accordance with the Declaration of Helsinki and the recommendations for Good Clinical Practice. Written informed consent will be signed by all study participants before enrolment in the study. Patients will have the right to withdraw from the study without incurring any prejudice at any time.

## Patient involvement

RM, first author and expert patient, and members of DUP GA participated in the design of this study and will participate in all stages including teaching peers[23] and promoting and reporting the data, including publication in peer review. Thanks to training with health professionals and expert patients[22 23 25] peers will adopt the appropriate posture to enable patients to find their own resources to overcome barriers to use CPAP.

## Dissemination

Dissemination plans of the results include presentations at conferences and a publication in peer-reviewed journal. Updates of the randomised trial will be available at ClinicalTrials.gov. All patients will be informed that the dissemination of results will be accessible on request.

## Sponsor and funding

The study sponsor will be AGIR à dom. Coprincipal investigators are RM, an expert patient and JCB, a researcher. The collaborators and sponsors were not involved in the design of the study and will not influence the execution, analysis or publication of results.

## DISCUSSION

OSAS is associated with many negative health consequences.[1] The lack of compliance with home CPAP therapy, which is the first line of treatment, and which has shown to be effective on quality of life is a major issue both in terms of the patient's own health status and in healthcare utilisation.[1 2 7 8] Attempts have been made to improve CPAP compliance by improving technical issues relating to the comfort of use of the system[10 11] and the use of the of remote monitoring and telemedicine, along with the implementation of web-based adherence interventions[12–15]; however, they have not been shown to improve compliance with the therapy. Other strategies to improve compliance therefore need to be developed and tested.

One of the main strengths of this study is the involvement of peers in the implementation of the behavioural intervention. Regarding efficacy, the involvement of patients with experience in the motivation of their peers to comply with treatment has been implemented with success in other chronic conditions requiring self-management such as HIV and diabetes.[33 34] Furthermore, evidence suggests that patients perceive peers with similar comorbidities as more credible than healthcare professionals in the delivery of behavioural interventions.[35–37] The concept of PPI in education and research has been adopted by a growing number of medical schools, particularly in the UK.[19 24] If the results of this study confirm the effectiveness of the PI intervention in promoting resumption of CPAP in patients initially failing CPAP, this study

will provide an evidence base to support the use of PI in the management of OSAS in conjunction with the home healthcare provider and specialised sleep centres.[38]

The aim to seek factors that are related to CPAP resumption will provide useful information regarding those patients who are more likely to resume CPAP and therefore who PI interventions are more likely to help. This will open the way for further studies to determine the most appropriate methods to improve compliance in those patients who benefit less from PI interventions.

Despite these strengths, the study has two main inherent limitations. First, the results are likely to be biased by the fact that patients who accept to participate may be more likely to resume CPAP therapy than those who decline participation. The results may therefore not be generalisable to all patients who have stopped using their CPAP as prescribed. Second, the effectiveness of the intervention may also depend on the capacity of the peer-participant to deliver it. The training is quite short (three half-days) and some of the peers recruited may be more skilled than others in providing such intervention. However, in this study, the peers will be additionally supported throughout the study by the DUP.

In summary, the results of this study will determine the effectiveness of a PI intervention to motivate patients who have stopped using their CPAP as prescribed to resume its use on compliance with CPAP therapy. The results will also provide information regarding the factors relating to resumption of CPAP, providing a starting point for further studies to determine the most appropriate methods to improve compliance in those patients who benefit less from PI interventions.

**Author affiliations**
[1]Université Grenoble Alpes, Saint-Martin-d'Heres, France
[2]Service Hospitalier Universitaire Pneumologie Physiologie, Pôle Thorax et Vaisseaux, Centre Hospitalier Universitaire Grenoble Alpes, Grenoble, France
[3]Agir à dom, Meylan, France
[4]Centre Hospitalier Universitaire Grenoble Alpes, Grenoble Alpes, France
[5]HP2; Inserm, U1042, Univ. Grenoble Alpes, Grenoble, France

**Acknowledgements** We specially thank John Louis McGregor, PhD, retired Director of Medical Research (DR1) at INSERM, former director of INSERM Unit 331, and retired Honorary Senior Lecturer (research) at the Cardiovascular division King's College London, for his continuous encouragements and counselling together with manuscript reviewing.

**Contributors** RM participated in the design of the study, wrote the article based on the study protocol, will train PI, collect and analyse data into the protocol. CP participated in the design of the study, wrote the study protocol and will include patients into the protocol together with PPI. SL participated in the design of the study, wrote the study protocol. CD and NA participated in establishing the sample size and will help to recruit patients. MR set up statistical analysis plan and determine sample size. RT revised the manuscript, will include patients into the protocol and collect and analyse data. JLP designed the study, critically revised the manuscript, will include patients and collect and analyse data. JCB designed the study, critically revised the manuscript and will analyse data. The submitted manuscript has been approved by all authors.

**Funding** RM is a recipient of a grant from Agir pour les Maladies Chroniques, http://fonds-apmc.org received in 2019. JLP and RT are supported by the French National Research Agency in the framework of the 'Investissements d'avenir' programme (ANR-15-IDEX-02) and the 'e-health and integrated care and trajectories medicine and MIAI artificial intelligence' Chairs of excellence from the Grenoble-Alpes University Foundation. This work has been partially supported by MIAI @ Grenoble Alpes, (ANR-19-P3IA-0003).

**Competing interests** RM is a recipient of a grant from Agir pour les Maladies Chroniques, http://fonds-apmc.org/. CD, NA, JCB are employees of AGIR à dom. CP and JLP received grants from Agir pour les Maladies Chroniques, http://fonds-apmc.org/.

**Patient and public involvement** Patients and/or the public were involved in the design, or conduct, or reporting, or dissemination plans of this research. Refer to the Methods section for further details.

**Patient consent for publication** Consent obtained directly from patient(s)

**Provenance and peer review** Not commissioned; externally peer reviewed.

**ORCID iDs**
Christophe Pison http://orcid.org/0000-0002-2152-6461
Matthieu Roustit http://orcid.org/0000-0003-4475-1626
Jean Christian Borel http://orcid.org/0000-0003-4140-6210

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
