## [Reviewer comments · BMJ Open]

ARTICLE DETAILS

TITLE (PROVISIONAL)	A peer-driven intervention to help patients resume CPAP therapy following discontinuation: a multicenter, randomized clinical trial with patient involvement
AUTHORS	

VERSION 1 – REVIEW

REVIEWER	Phillips, Craig L. Univ Sydney
REVIEW RETURNED	05-Jul-2021

GENERAL COMMENTS	The manuscript is well written. The study rationale is sound and the trial, if completed will provide valuable information about CPAP abandonment after longer-term use in severe OSA. The investigators take a different approach by involving consumers in this study which is encouraging to see. I only have a few comments. As a non-French speaker, I was unable to review the trial Protocol as it was written in French. Please clarify whether it was prepared in accordance with the SPIRIT checklist? https://www.spirit-statement.org. If not, then I encourage the authors refer to this document and ensure that all items in the checklist have been followed as it will complement the CONSORT statement when reporting results. For example, I could not tell what blinding measures are to be used as listed in 17a: Who will be blinded after assignment to interventions (eg, trial participants, care providers, outcome assessors, data analysts) and how. Page 3 – Strengths and limitations of the study. This list is more related to the study rationale and feasibility. For example, an unmet need provides rationale for conducting the study but is not necessarily a strength. Having experience in PPI provides evidence of feasibility whereas a study design that incorporates PPI could be considered a strength. I would suggest compiling this list of strengths and weaknesses according to those raised in the discussion (e.g. PPI in the study design, participation bias and, heterogeneity in the effectiveness of the intervention amongst peer participants). Page 14: “Updates of the randomized trial will be available at ClinicalTrials.com.” Is this not ClinicalTrials.gov?
---

REVIEWER	Xia, Yunyan Department of otolaryngology head & neck surgery of the First Affiliated Hospital of Nanchang University, Department of
-----------------	--

	Otolaryngology Head and Neck Surgery & Center of Sleep Medicine
REVIEW RETURNED	06-Jul-2021

GENERAL COMMENTS	This paper can be published after making some minor revision described below:  1. Will it be too many for a PI to be allocated to 5 to 8 patients? 2. Will you screen the patients for the reasons of stopping CPAP? As for some patients, they might stop the CPAP treatment for some problem that are easy to be solved, which might affect the results.
--

VERSION 1 – AUTHOR RESPONSE

Reviewer: 1, Dr. Craig L. Phillips, Univ Sydney

Comments to the Author: The manuscript is well written. The study rationale is sound and the trial, if completed will provide valuable information about CPAP abandonment after longer-term use in severe OSA. The investigators take a different approach by involving consumers in this study which is encouraging to see. I only have a few comments.

As a non-French speaker, I was unable to review the trial Protocol as it was written in French. Please clarify whether it was prepared in accordance with the SPIRIT checklist? <https://www.spirit-statement.org>. If not, then I encourage the authors refer to this document and ensure that all items in the checklist have been followed as it will complement the CONSORT statement when reporting results.

We added a SPIRIT checklist.

For example, I could not tell what blinding measures are to be used as listed in 17a: Who will be blinded after assignment to interventions (eg, trial participants, care providers, outcome assessors, data analysts) and how.

Outcome assessors will be blinded.

Page 3 – Strengths and limitations of the study. This list is more related to the study rationale and feasibility. For example, an unmet need provides rationale for conducting the study but is not necessarily a strength. Having experience in PPI provides evidence of feasibility whereas a study design that incorporates PPI could be considered a strength. I would suggest compiling this list of strengths and weaknesses according to those raised in the discussion (e.g. PPI in the study design, participation bias and, heterogeneity in the effectiveness of the intervention amongst peer participants).

We agree to follow for this section the points raised along the discussion.

Page 14: “Updates of the randomized trial will be available at ClinicalTrials.com.” Is this not ClinicalTrials.gov?

You are right.

Reviewer: 2 Dr. Yunyan Xia, Department of otolaryngology head & neck surgery of the First Affiliated Hospital of Nanchang University

Comments to the Author:

This paper can be published after making some minor revision described below:

1. Will it be too many for a PI to be allocated to 5 to 8 patients?

Each PI along the trial could follow at maximum 5 to 8 patients in a one-to-one meeting on 3 occasions; we anticipate to train more PI if it appeared it is a too large burden.

2. Will you screen the patients for the reasons of stopping CPAP? As for some patients, they might stop the CPAP treatment for some problem that are easy to be solved, which might affect the results.

You are totally right; patients will be only included after failure of providers and doctors to resume CPAP treatment with the machine returned to the provider.

VERSION 2 – REVIEW

REVIEWER	Phillips, Craig L. Univ Sydney
REVIEW RETURNED	19-Aug-2021

GENERAL COMMENTS	The authors have addressed all my points.
---

REVIEWER	Xia, Yunyan Department of otolaryngology head & neck surgery of the First Affiliated Hospital of Nanchang University, Department of Otolaryngology Head and Neck Surgery & Center of Sleep Medicine
REVIEW RETURNED	15-Aug-2021

GENERAL COMMENTS	This is a well written study protocol. It would be better if you described more about the inclusion criteria of the patients.
--

VERSION 2 – AUTHOR RESPONSE

Reviewer: 2

This is a well written study protocol. It would be better if you described more about the inclusion criteria of the patients.

We recapitulate these key informations in manuscript page 8, lines 184-188 and in Table 1.

We corrected reference citation order, added in-text citation for Table 1 and moved Supplementary Checklist in a Supplementary file,

We hope it will now meet the standards of BMJ Open protocol.